# Adaptation of the infant gut microbiome during the complementary feeding transition

**Starin McKeen**[1,2,3,4], **Nicole C. Roy**[1,4,5], **Jane Adair Mullaney**[1,3,4], **Hannah Eriksen**[4,6], **Amy Lovell**[4,6], **Martin Kussman**[7], **Wayne Young**[1,3,4], **Karl Fraser**[1,3,4], **Clare R. Wall**[4,6], **Warren C. McNabb**[1,4]*

**1** Riddet Institute, Massey University, Palmerston North, New Zealand, **2** School of Food and Advanced Technology, Massey University, Palmerston North, New Zealand, **3** AgResearch Ltd, Palmerston North, New Zealand, **4** High-Value Nutrition National Science Challenge, Auckland, New Zealand, **5** Department of Human Nutrition, University of Otago, Dunedin, New Zealand, **6** Department of Nutrition and Dietetics, The University of Auckland, Auckland, New Zealand, **7** German Entrepreneurship, Cambridge, Massachusetts, United States of America

* W.McNabb@massey.ac.nz

**Editor:** yinglin xia, University of Illinois at Chicago College of Medicine, UNITED STATES

**Data Availability Statement:** All the data has been uploaded as supplemental information (S3-S45 in a zip file).

## Abstract

The infant gut microbiome progresses in composition and function during the introduction of solid foods throughout the first year of life. The purpose of this study was to characterize changes in healthy infant gut microbiome composition, metagenomic functional capacity, and associated metabolites over the course of the complementary feeding period. Fecal samples were obtained at three 'snapshot' timepoints from infants participating in the 'Nourish to Flourish' pilot study: before the introduction of solid foods at approximately 4 months of age, after introducing solid foods at 9 months of age, and after continued diet diversification at 12 months of age. KEGG and taxonomy assignments were correlated with LC-MS metabolomic profiles to identify patterns of co-abundance. The composition of the microbiome diversified during the first year of life, while the functional capacity present in the gut microbiome remained stable. The introduction of solid foods between 4 and 9 months of age corresponded to a larger magnitude of change in relative abundance of sequences assigned to KEGG pathways and taxonomic assignments, as well as to stronger correlations with metabolites, compared to the magnitude of changes and number of correlations seen during continued diet diversification between 9 and 12 months of age. Changes in aqueous fecal metabolites were more strongly correlated with KEGG pathway assignments, while changes in lipid metabolites associated with taxonomic assignments, particularly between 9 and 12 months of age. This study establishes trends in microbiome composition and functional capacity occurring during the complementary feeding period and identifies potential metabolite targets for future investigations.

## Introduction

The infant gut microbiome undergoes dramatic shifts in composition from the unstable neonatal microbiome towards a more stable microbiome throughout the first year of life [1]. This

**Funding:** SM was funded for a PhD scholarship for this study by the New Zealand Ministry of Business Innovation and Employment as part of the High Value Nutrition National Science Challenge. https://www.mbie.govt.nz/science-and-technology/science-and-innovation/funding-information-and-opportunities/investment-funds/national-science-challenges/ The funders had no role in study design, data collection and analysis, decision to publish, or preparation of the manuscript.

**Competing interests:** The authors have declared that no competing interests exist.

progression has been characterised as non-random in response to life events and environmental exposures, such as diet [2]. Because the introduction of solid foods is a disruptive event for the infant gut ecosystem, it provides an opportunity to understand how the early infant gut microbiome adapts in both composition and function over time [3]. It is accepted that the composition of microbial species diversifies throughout the first year of life. However, it is less clear how the functional capacity of the microbiome changes in response to complementary feeding: the addition of solid foods to the milk-based diet.

At the phylum level, the gut microbiome of neonatal infants is dominated by approximately 70% Actinobacteria and Proteobacteria and approximately 30% Firmicutes, Bacteroidetes, and Verrucomicrobia [4]. By adulthood, the microbiome is composed of approximately 90% Firmicutes and Bacteroidetes, and approximately 10% Actinobacteria, Proteobacteria, and Verrucomicrobia. The majority of this transition occurs within the first two to three years of life [5].

Actinobacteria are efficient utilizers of oligosaccharides, one of the primary components of breastmilk, and thrive in the infant gut [6]. Proteobacteria and Bacteroidetes utilize proteins and saccharides, substantial components of breastmilk and infant formula, which may allow members of these families to persist throughout the transition to diverse substrates from solid food [7]. The Firmicutes phylum encompasses a diversity of families and species of microbes that collectively provide the enzymatic capability to utilize carbohydrates, proteins, saccharides, and fermentative by-products of other microbes such as formate, lactate, succinate, acetate, and various gases [8]. This metabolic diversity allows Firmicutes to fill new resource niches that are created during dietary diversification. Collectively, the gut microbiome of milk-fed infants adapts to a rapidly changing diet during the complementary feeding period through shifts in populations of microbial species with a vast enzymatic capacity.

This is a secondary analysis of a pilot study. The pilot study was designed to pilot the methodologies required for a randomized control trial (RCT). In this study we hypothesised that changes in the composition, rather than function, of the gut microbiome of infants, are more significant during the time-period when infants are introduced to solid foods (between 4 and 9 months of age), compared to approximately 3 months later (between 9 and 12 months of age), when the diet has continued to diversify. Shotgun metagenomics and Liquid Chromatography-Mass Spectrometry (LC-MS) based metabolomics were carried out on fecal samples collected from infants at three different time points to characterise changes in the microbiome and metabolome following a period of dietary diversification. The 'snapshot' time points were at 4.1 months of age ± 31 days (before the introduction of solid foods), at 8.7 months of age ± 18 days, and at 11.7 months of age ± 18 days (subsequently referred to as 4, 9, and 12 months of age respectively). Sequences assigned to taxonomy and gene functions (KEGG pathways), and metabolomic data were integrated into a longitudinal analysis to identify key associated features.

## Materials and methods

### Ethics approval and consent to participate

Ethics approvals were granted by the New Zealand Health and Disability Ethics Committee (Ref: 17/NTA/239). Written informed consent was obtained from parents on the infant's behalf.

Data collection and analysis will not be performed blind to the conditions of the experiments. Forty infants who primarily consumed breastmilk and had not yet begun consuming solid foods were recruited from the Auckland Metropolitan area between January and December 2018. The participants were part of the 'Nourish to Flourish trial', a pilot study designed to assess the feasibility of conducting nutrition research and systems biology analyses

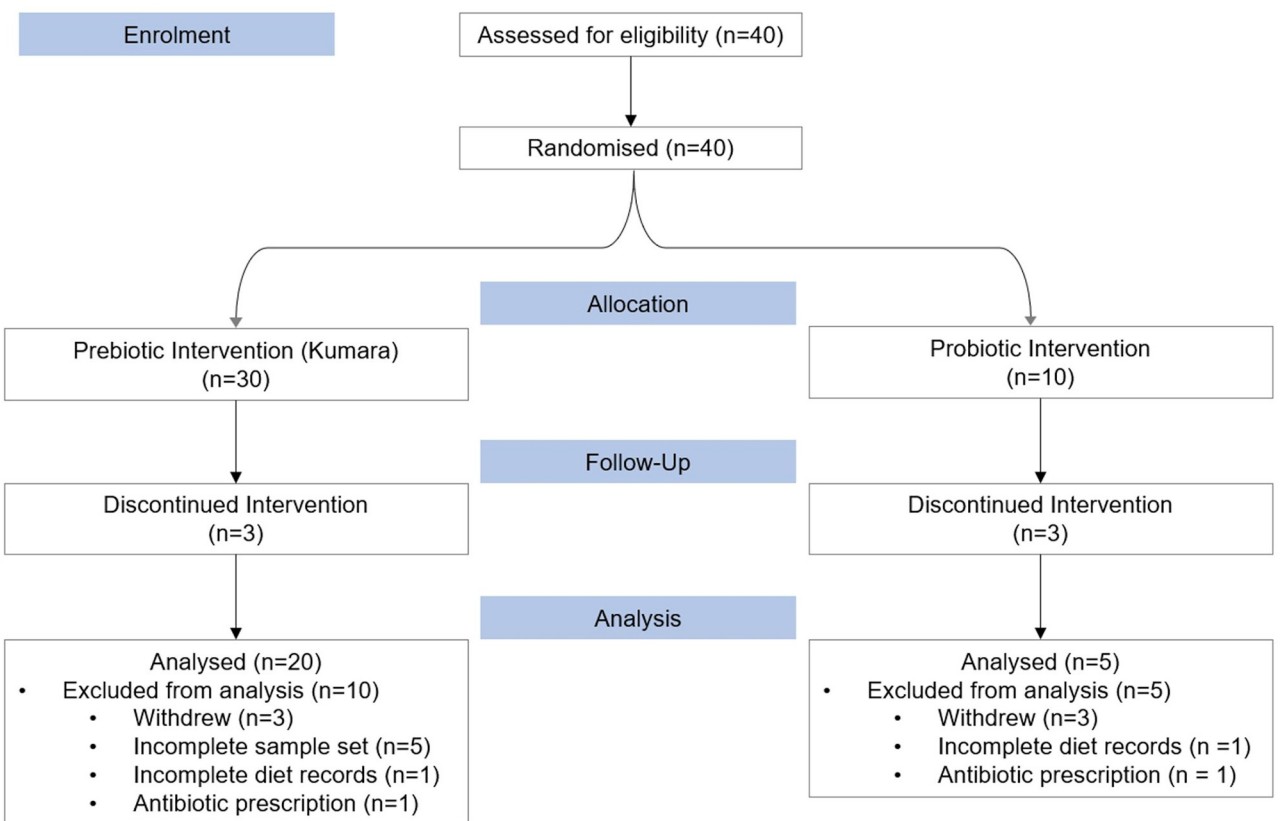

**Fig 1. CONSORT flow of Nourish to Flourish pilot study (ANZ Clinical Trials Registry number 12618000157279).**

in weaning infants. Trial registration: Australia New Zealand Clinical Trials Registry, ANZCTR12618000157279. Registered February 1, 2018, http://www.anzctr.org.au/. Infants were randomised by numbers 1–10 and 11–40 into either of two intervention groups, one consuming a daily dose of commercially available probiotics (1 billion CFU of *Bifidobacterium animalis* ssp *lactis*) (n = 10), and the other consuming 5 grams of lyophilised kumara (a variety of New Zealand sweet potato, including the skin) daily (n = 30) (Fig 1). Throughout the study six infants withdrew, five were excluded from the analysis for incomplete sample sets, two were excluded from analysis for incomplete diet data records, and two were excluded due to oral antibiotic prescription within 21 days of sample collection (Fig 1). The remaining 25 infants were included in the subsequent analyses presented here.

## Sample collection

Stool samples were collected by parents from nappies during the three days before clinic visits. At the 4 months of age clinic visit, before the introduction of solid foods, a nappy liner was provided to accommodate the liquid consistency of infant stools. Samples at later time points were scooped into collection pottles. Parents were asked to store samples in the home freezer (-20˚C) and were provided with Styrofoam boxes and ice packs for transport to the clinic. Stool samples were subsequently stored at -80˚C until processed for microbiome and metabolome analyses (Fig 2). Analyses were conducted in two batches: batch 1 included samples from all 25 infants at 4 months of age, approximately half of the 9 month old samples, and 6 of the

12 month old samples. Batch 2 included the remaining 9 and 12 month old samples as well as a repeated 4 month old sample from batch 1 for quality control.

## Fecal microbiome

A single fecal aliquot was vortexed, centrifuged at 3,000 rcf for 15 min to separate the bacterial pellet from the supernatant, and the supernatant was removed. This process was repeated for low volume samples to achieve approximately 100 mg to carry out DNA isolations DNA was isolated from fecal samples using a Macherey-Nagel DNA isolation kit, quantified using a Nanodrop Spectrophotometer (Thermo Fisher Scientific), and the approximate degree of degradation and fragmentation of DNA was visualised by gel electrophoresis.. The APC Microbiome Institute, Teagasc, part of the Agricultural and Food Development Authority in Cork, Ireland carried out the shotgun metagenomic sequencing using Illumina NextSeq paired-end 150 bp x 2 sequencing. Libraries were prepared using the Illumina Nextera XT kit. Paired sequences were joined with PEAR version 0.9.6 [9]. Host sequences were detected and removed using the bbduk.sh function from the BBMAP package version 38.22–0 [PM2], with the human genome (Human GRCh38) as reference. Metaxa2 version 2.1.3a [10] was used to identify SSU ribosomal DNA and taxonomy was assigned using the Silva 128 database [11]. The "blastx" function of DIAMOND version 0.9.22 [12] was used to map the reads against the "nr" NCBI database. Megan version 6 ultimate edition [13] was used to assign putative functions to the DIAMOND alignment files against the KEGG database [14–16].

Phylum and species-level taxonomic compositions were used for downstream analyses (Fig 2). Composition tables were filtered to exclude rare species, defined here as species present

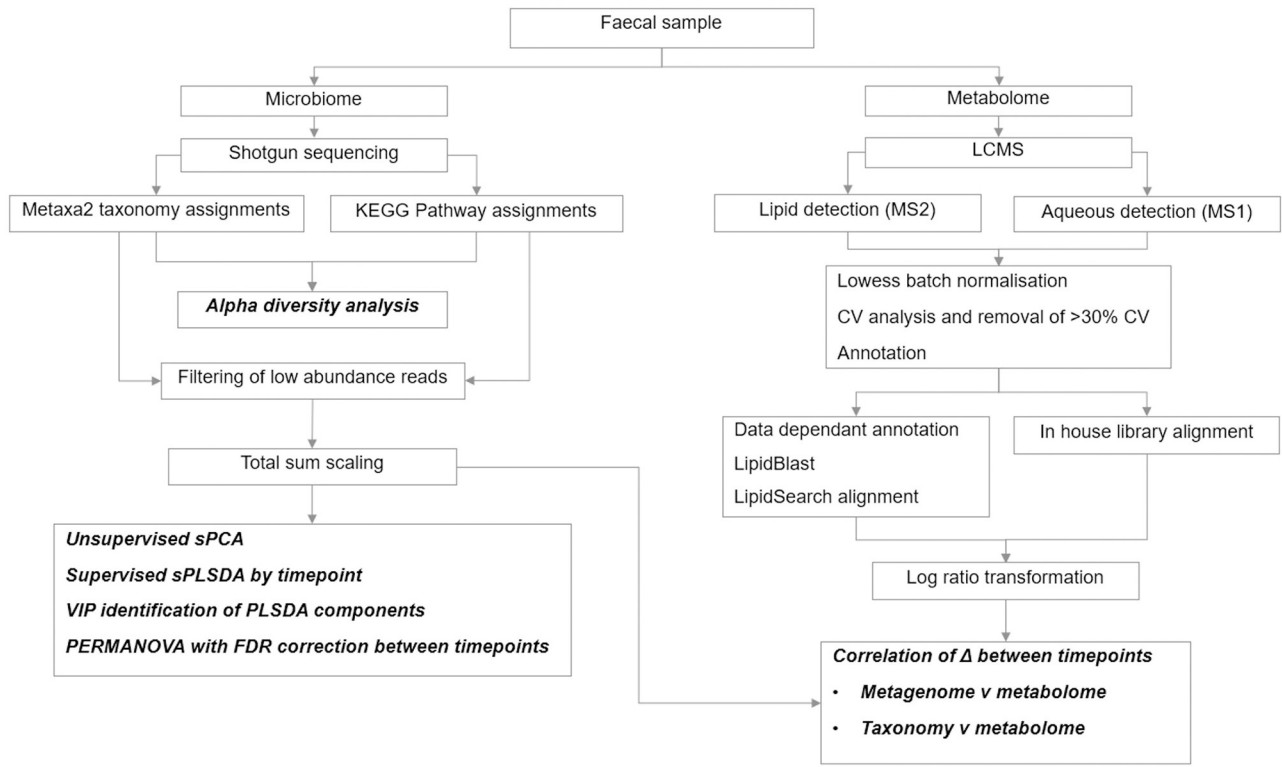

**Fig 2. Diagram of the analytical approach for elucidating the shifts in microbiome composition and function over time in the Nourish to Flourish cohort.** Specific analyses are in bold italic.

at < .001% in fewer than three samples. KEGG pathways are structured into levels with level 1 grouped into seven; metabolism, genetic information processing, cellular processes, organismal systems, human diseases and drug development. Level 2 further stratifies the grouping with each having several sub groupings down to level 3 which then identifies the specific gene pathway involved (glycan biosynthesis for example). A total of 116 level 6 Metaxa2 assignments and a total of 184 KEGG pathway assignments (Level 3) were included in subsequent analyses. Total sum scaling within individual samples was used to indicate relative abundances of species and KEGG pathways.

## Fecal metabolome

The normalisation of fecal samples by weight was carried out by lyophilising and homogenising samples. Biphasic extraction of 50 mg fecal sample was performed with methanol:water: methyl tert-butyl ether (6:7:10v/v). An aliquot of the methanol:water phase containing the aqueous metabolites, and an aliquot of the methyl tert-butyl ether phase containing the lipids, were subsequently dried on a heating block (30˚ C) under a continuous stream of nitrogen. Lipid extracts were reconstituted with chloroform:methanol (2:1 v/v) with a deuterated phosphatidylethanolamine internal standard. Aqueous samples were reconstituted with acetonitrile:water (1:1 v/v) with deuterated amino acid internal standards. Instrument methods, including columns, solvents, gradients, and spray voltage, were the same for both LC-MS analysis batches. LC-MS metabolomics was performed using a high-resolution Orbitrap (Thermo) system, with polar metabolites separated using HILIC chromatography and lipids separated using a CSH-C18 reversed-phase separation [17–19]. Both polar metabolite and lipidomic LC-MS analyses were performed in both positive and negative ionisation modes.

The MSConvert function of ProteoWizard™ was used to convert Thermo RAW data files to mzML files. Peak detection, retention time alignment, grouping and gap filling were carried out using XCMS [20] and in-house scripts in R version 3.6.2 [21]. The 'diffreport' function of XCMS was used to carry out a two-group comparison between (1) pooled quality control (QC) samples from Batch 1 and blank samples and (2) QC and participant samples. The QC vs blanks comparison was used to identify non-significant (α = 0.05) $m/z$ features that correspond to background noise contributed by blanks, which were subsequently removed from the QC vs samples list of features. The feature list generated by comparing QC vs samples was further cleaned by browsing extracted ion chromatograms generated by the diffreport function in XCMS and deleting low quality $m/z$ features that represented background noise. The filtered data matrices were then subjected to Lowess normalisation using Workflow4Metabolomics (Galaxy) and subsequent filtering based on the coefficient of variance (>30% CV removed).

The resultant data matrix was used for downstream statistical analyses and identification by LipidSearch™, based on comparison of the ddMS$^2$ data obtained from selected samples from *in silico* calculations and database. The Q-Exactive database with the product search option was selected. Default parameters were used for LipidSearch™ quantitation. For this study, [M-H]$^-$ and [M+HCOO]$^-$ adducts, and [M+H]$^+$, [M+NH$_4$]$^+$, [M+Na]$^+$ adducts were selected for negative and positive ionisation modes, respectively. A retention time tolerance of 0.01 min and mass error tolerance of ± 5 ppm was allowed. Results and identifications from individual MS$^2$ files (LipidSearch™ results) were concatenated into a single table (one each for positive and negative ionisation modes) and matched with the respective, normalised, peak intensity table (XCMS results). Identification of water-soluble (aqueous) metabolites was carried out by aligning unique $m/z$ and retention times with an in-house library of metabolites specific to the instruments and columns used at AgResearch, which has been compiled and tailored from

external reference libraries. Concentrations of metabolites were log-transformed for subsequent analyses.

## Statistical and bioinformatic analyses

Statistical analyses were carried out in R version 4.0.2 on total sum scaled taxonomy and metagenome datasets, and log-transformed metabolome data. These methods maintain positive values, which is critical when assessing the direction of change in abundance or concentration. Ordination analyses, including sparse Principle Component Analysis (sPCA) and sparse Partial Least Squared Discriminant Analysis (sPLSDA) and regularised Canonical Correlation, were carried out using the MixOmics package version 6.1.1 in R version 4.0.2 after filtering low-abundance reads and total sum scaling [22]. Further identification of variables important to projection (VIPs) was identified using MixOmics, and a variable's scores from each component were combined into a sum VIP score to better understand the contribution of a variable to the given model. Regularised canonical correlations were based on Pearson's correlations of the delta (change) between time points and were carried out on 3 components using the clustered image heatmap function in MixOmics before filtering significant correlation coefficients from the resulting matrix. A correlation cut-off of 0.6 was used for the KEGG pathway correlations, whereas a correlation cut-off of 0.5 was used for species correlations, based on differences in the significance of correlations between the given datasets. Permutational analysis of variance (PERMANOVA) was carried out in the RVAideMemoire package version 0.9–7.8 at 2000 permutations per time point comparison, and Benjamin-Hochberg false discovery rate (FDR) corrected p-values used to determine significance. Non parametric one way ANOVA was used to identify whether groups were different and then this was followed up with post hoc testing (Tukeys).

## Results

As a pilot investigation, the Nourish to Flourish study demonstrated the safety and feasibility of conducting nutritional systems biology research in infants during complementary feeding. Results presented here will focus on the changes in microbial composition and predicted functional capacity over the complementary feeding period, and associated changes in metabolite concentrations.

### Broad trends in the gut microbiome composition and function

Before the introduction of solid foods, Actinobacteria, such as *Bifidobacterium*, were the most abundant (Fig 3A). Firmicutes gradually increased in abundance from approximately 25% at 4 months of age to approximately 53% of sequences assigned to phyla at 12 months of age (Fig 3A). Both Actinobacteria and Proteobacteria decreased in relative abundance (by 10% and 20% respectively), and members of the Bacteroidetes phylum increase by 8% over time (Fig 3A). Species belonging to Verrucomicrobia, Fusobacteria, Cyanobacteria, Saccharibacteria and Tenericutes phyla were present in much lower proportions at all time points (Fig 3A).

Metagenomic sequences were predominantly annotated to "signalling and cellular processes" KEGG pathways at Level 2 classification at 4 months of age and decreased over time from approximately 19.4% to 17.1% of total sequences annotated (Fig 3B). Genes assigned to "genetic information processing" increased from approximately 15.1% at 4 months of age to 17.2% by 12 months of age (Fig 3B). Sequences assigned to "metabolism of carbohydrates", "proteins" and "amino acids" remained at approximately 9%, 8%, and 6%, respectively, of total sequences annotated to level 2 KEGG pathway classifications over time (Fig 3B). The variation in the relative abundance of sequences assigned to level 2 KEGG pathways was smaller than

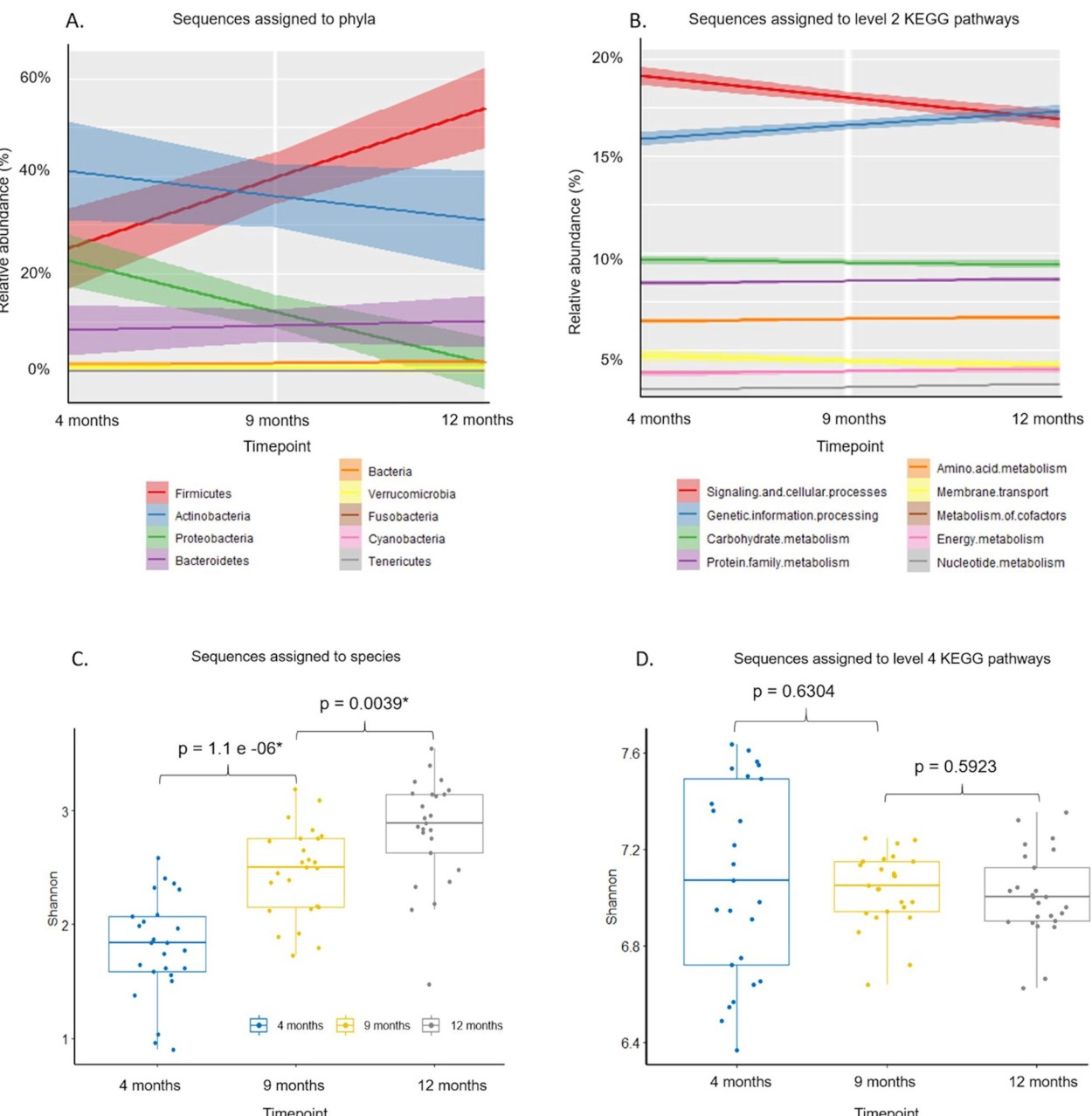

**Fig 3. Panels A and B show trends in sequences assigned to phyla and level 2 KEGG pathways at 4 months of age, 9 months of age, and 12 months of age.** Panels C and D show Shannon alpha diversity index at each time point with significant differences evaluated by post hoc testing between sampling time points for sequences assigned to level 7 Metaxa2 species and sequences assigned to level 4 KEGG pathways, respectively. The 4 month old time point is shown in blue, 9 months in orange and 12 months in grey. The shading around the lines in A and B indicate minimum—maximum values.

the differences in the relative abundance of sequences assigned to dominant phyla, shown by the shaded areas around trend lines, and the relative abundances were smaller for functional groups than for phyla (Fig 3A and 3B).

The diversity of species in the infant gut microbiome increased significantly with the addition of solid foods to the milk-based diet (Fig 3C). However, the diversity of functional genes

remained, on average, the same, as measured by Shannon alpha diversity index in the R package "vegan" at level 7 of Metaxa 2 taxonomic assignments and level 4 KEGG pathway assignments (Fig 3D). The increase in taxonomic alpha diversity was more significant between 4 and 9 months of age than between 9 and 12 months of age (Fig 3C). The alpha diversity scores for relative abundances of sequences assigned to KEGG pathways were more variable at 4 months of age, but the difference narrowed at 9 and 12 months of age (Fig 3D).

## Shifts in the fecal microbiome and metabolome between time points

Both sPCA and sPLS-DA were carried out on genus level Metaxa2 assignments, level 3 KEGG pathway assignments, and both aqueous and lipid metabolites to visualise the changes in the infant gut microbiome throughout the complementary feeding period (Fig 4).

The sPCA did not show any obvious separation between groups. Application of sPLS-DA, a supervised ordination analysis, also did not increase the separation between time points, although the overlap between time point clusters was reduced in the sPLS-DA models of sequences assigned to KEGG pathways, aqueous metabolites, and lipid metabolites (Fig 4D). The percentage of explained variance was lower at 12 months of age than 9 months of age for Metaxa2 assignments, KEGG pathway assignments, and metabolites (Fig 4). KEGG pathway assignments had the highest percentage of explained variance at both timepoints, followed by aqueous metabolites (Fig 4).

The taxonomic composition was most similar between individual infants at 9 months of age, compared to at 4 or 12 months of age (Fig 4A). Sequences assigned to KEGG pathways became increasingly similar between individuals over time (Fig 4C), in agreement with the alpha diversity patterns shown in Fig 3D. Sequences assigned to Metaxa2 classifications and KEGG pathways separated into two distinct clusters at 4 months of age in both sPCA and sPLS-DA models (Fig 4C and 4D). This separation was predominantly associated with mode of birth (i.e. caesarean section vs vaginal delivery), but the influence of other factors such as early-life environmental, genetic, and dietary factors could not be ruled out. The differences between these infants diminished at 9 and 12 months of age.

Metabolite profiles were more variable between infants at all time points than microbiome taxonomy or KEGG pathway assignments, as shown by the scales on which the sPCA ordinations were cast (Fig 4). Supervision by time point in sPLS-DA models led to the separation of the 4 months of age samples, but the 9 and 12 months of age metabolite profiles were similar in these models (Fig 4F and 4G). Aqueous metabolite profiles became more similar between infants at the 9 month time point, compared to infants at either 4 or 12 months of age (Fig 4E). Lipid metabolite profiles of 4 month old samples were similar among infants in the sPCA model (Fig 4G). These similarities diminished at 9 and 12 months (Fig 4G).

Separate sPLS-DA models were carried out to compare taxonomy assignments and KEGG pathway assignments between 4 and 9 months of age and 9 and 12 months of age. VIPs important to sPLS-DA were selected from these two-time point models and importance was derived by adding variables' VIP scores from each of the three components, thus indicating how that variable contributed to the separation of each time point. PERMANOVA was used to compare the significance of the difference between the means and spread of relative abundance of individual taxonomic assignments and individual sequences assigned to KEGG pathways between time points. Variables considered significantly different by PERMANOVA, or which constituted a substantial proportion of the overall abundance, are shown in Figs 5 and 6 along with cumulative VIP scores, relative abundances, and log2 fold change between time points. Mean relative abundances of 0 for any given time point were imputed with a value $10^{-6}$. Considering

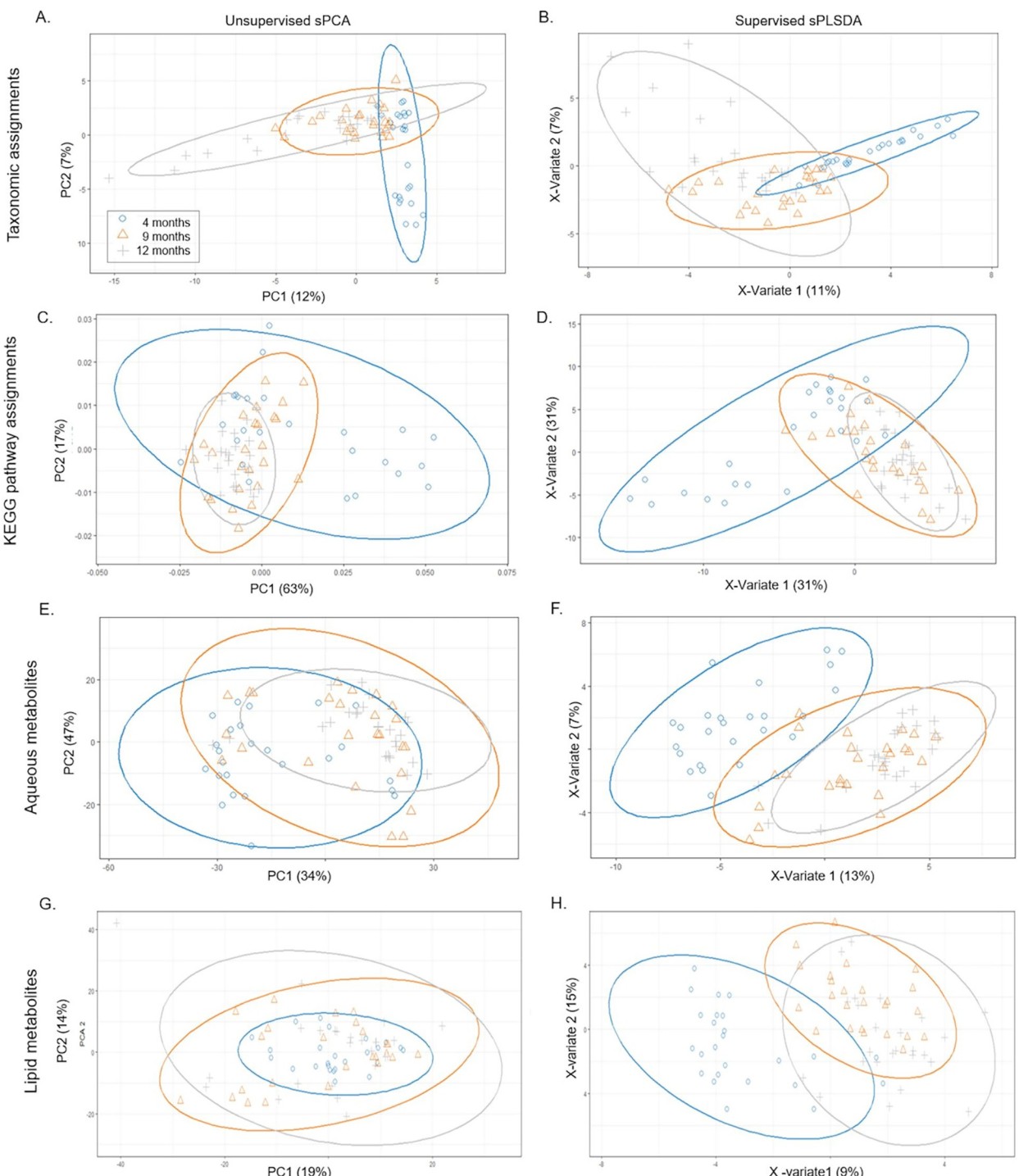

**Fig 4.** Ordination analyses of taxonomic assignments are shown in panels A and B, KEGG pathway assignments are shown in panels C and D, aqueous metabolites in panels E and F, and lipid metabolites in panels G and H. Sparse Principle Component Analyses are shown in the left hand column (A, C, E, and G) and sparse Partial Least Squared Discriminant Analysis plots supervised by time point, and cast on 3 components are shown in the right hand column (B, D, F, and H). Four month old samples are depicted in blue, 9 month old samples in orange, and 12 month old samples in grey with individuals marked in circles, triangles, and crosses respectively. Ellipses depict 95% confidence of clustered samples. The explained variance (%) is listed in parenthesis on each axis.

| Phylum | Taxa | Relative abundance (%) | | | 4 vs 9 months | | | 9 vs 12 months | | | Trend |
|---|---|---|---|---|---|---|---|---|---|---|---|
| | | 4 months | 9 months | 12 months | Log2 FC | VIP Sum | fdr p-value | Log2FC | VIP Sum | fdr p-value | |
| Actinobacteria | *Bifidobacteriaceae Bifidobacterium longum* | 0.55% | 0.33% | 0.19% | -0.73 | 2.71 | 0.04 | -0.80 | 3.38 | 0.04 | |
| | *Bifidobacteriales* | 0.02% | 0.01% | 0.00% | -0.96 | 3.93 | 0.01 | -1.69 | 3.13 | 0.01 | |
| | *Bifidobacteriaceae*** | 36.82% | 30.68% | 26.79% | -0.26 | 2.18 | 0.50 | -0.20 | 0.31 | 0.51 | |
| Firmicutes | *Ruminococcaceae Subdoligranulum* | 0.00% | 0.31% | 0.96% | 8.19 | 1.27 | 0.01 | 1.63 | 3.57 | 0.01 | |
| | *Ruminococcaceae Ruminococcus* | 0.00% | 0.10% | 0.47% | 5.73 | 1.22 | 0.01 | 2.17 | 3.45 | 0.02 | |
| | *Ruminococcaceae Faecalibacterium* | 0.01% | 1.11% | 3.62% | 6.66 | 1.84 | 0.00 | 1.70 | 6.60 | 0.00 | |
| | *Ruminococcaceae* | 0.18% | 0.77% | 2.06% | 2.13 | 2.33 | 0.01 | 1.42 | 7.19 | 0.00 | |
| | *Ruminococcaceae Ruminococcaceae* | 0.01% | 0.03% | 0.16% | 1.86 | 0.17 | 0.01 | 2.67 | 4.39 | 0.01 | |
| | *Lachnospiraceae Tyzzerella* | 0.01% | 0.02% | 0.01% | 3.13 | 1.93 | 0.01 | 1.82 | 2.48 | 0.01 | |
| | *Lachnospiraceae Sellimonas* | 0.00% | 0.02% | 0.08% | 0.96 | 0.43 | 0.03 | 2.25 | 3.40 | 0.03 | |
| | *Lachnospiraceae Roseburia* | 0.01% | 0.04% | 0.19% | 7.42 | 1.41 | 0.04 | 0.78 | 0.57 | 0.03 | |
| | *Lachnospiraceae Lachnospiraceae* | 0.00% | 0.80% | 0.28% | 2.34 | 1.10 | 0.00 | 2.15 | 3.72 | 0.00 | |
| | *Lachnospiraceae Coprococcus* | 0.00% | 0.67% | 0.55% | 1.69 | 0.43 | 0.00 | 3.12 | 7.98 | 0.00 | |
| | *Lachnospiraceae Blautia* | 0.00% | 0.03% | 0.01% | 2.43 | 3.45 | 0.01 | 0.70 | 1.29 | 0.00 | |
| | *Lachnospiraceae* | 1.89% | 11.00% | 13.71% | 2.54 | 7.57 | 0.00 | 0.32 | 0.05 | 0.00 | |
| | *Erysipelotrichaceae Turicibacter* | 0.00% | 0.00% | 0.02% | 3.86 | 2.45 | 0.04 | 2.56 | 0.96 | 0.04 | |
| | *Unclassified Clostridia* | 0.01% | 0.06% | 0.08% | 2.44 | 8.64 | 0.00 | 0.40 | 0.79 | 0.00 | |
| | *Clostridiales* | 1.22% | 4.08% | 6.13% | 1.74 | 6.13 | 0.00 | 0.59 | 2.98 | 0.00 | |
| | *Bacillales* | 0.00% | 0.01% | 0.06% | 1.16 | 0.04 | 0.01 | 2.49 | 5.28 | 0.01 | |
| | *Peptostreptococcaceae Intestinibacter* | 0.00% | 0.06% | 0.07% | 4.78 | 10.27 | 0.01 | 0.15 | 1.93 | 0.00 | |
| | *Peptostreptococcaceae* | 0.06% | 0.38% | 0.37% | 2.75 | 5.04 | 0.04 | -0.07 | 1.67 | 0.04 | |
| | *Peptostreptococcaceae Peptoclostridium* | 0.02% | 0.16% | 0.12% | 3.14 | 5.07 | 0.03 | -0.43 | 1.89 | 0.01 | |
| | *Veillonellaceae* | 0.17% | 1.36% | 1.04% | 2.98 | 7.68 | 0.01 | -0.39 | 0.55 | 0.01 | |
| | *Firmicutes Unclassified* | 0.28% | 0.91% | 0.79% | 1.69 | 3.52 | 0.04 | -0.20 | 0.93 | 0.02 | |
| | *Clostridiaceae Clostridium sensu stricto* | 0.03% | 0.00% | 0.30% | -2.40 | 2.36 | 0.00 | -0.96 | 5.91 | 0.01 | |
| | *Clostridiaceae Clostridium neonatale* | 0.02% | 0.00% | 0.00% | -6.62 | 3.23 | 0.00 | -27.75 | 1.46 | 0.00 | |
| | *Ruminococcaceae Butyricicoccus* | 0.04% | 0.01% | 0.31% | -2.52 | 2.30 | 0.01 | 5.62 | 6.76 | 0.00 | |
| | *Erysipelotrichaceae Holdemanella* | 0.00% | 0.00% | 0.06% | -0.36 | 0.96 | 0.01 | 7.38 | 5.39 | 0.01 | |
| | *Erysipelotrichaceae Erysipelotrichaceae* | 0.00% | 0.00% | 0.15% | -0.67 | 0.35 | 0.00 | 6.12 | 4.60 | 0.00 | |
| | *Bacillaceae* | 0.00% | 0.00% | 0.02% | -2.68 | 0.12 | 0.01 | 7.93 | 5.71 | 0.01 | |
| | *Bacillaceae Bacillus* | 0.01% | 0.00% | 0.20% | -0.58 | 0.03 | 0.01 | 5.79 | 5.87 | 0.01 | |
| | *Christensenellaceae Christensenellaceae* | 0.00% | 0.00% | 0.10% | -1.50 | 1.47 | 0.00 | 6.91 | 6.25 | 0.01 | |
| Proteobacteria | *Enterobacteriaceae Enterobacteriaceae* | 7.21% | 2.10% | 1.09% | -1.77 | 2.84 | 0.00 | -1.17 | 0.88 | 0.01 | |
| | *Enterobacteriaceae Escherichia-Shigella* | 0.58% | 0.06% | 0.02% | -1.78 | 5.65 | 0.00 | -0.94 | 3.90 | 0.00 | |
| | *Enterobacteriaceae Salmonella enterica PA* | 0.08% | 0.02% | 0.01% | -2.07 | 6.53 | 0.00 | -1.75 | 6.41 | 0.00 | |
| | *Enterobacteriaceae* | 16.40% | 3.82% | 1.87% | -2.10 | 7.83 | 0.00 | -1.04 | 6.02 | 0.00 | |
| | *Enterobacteriales* | 0.05% | 0.00% | 0.00% | -3.47 | 7.68 | 0.00 | -0.28 | 0.55 | 0.00 | |
| | *Gammaproteobacteria Unclassified* | 0.84% | 0.18% | 0.13% | -2.19 | 8.24 | 0.00 | -0.51 | 1.27 | 0.00 | |
| | *Vibrionaceae Vibrio* | 0.02% | 0.01% | 0.01% | -1.26 | 2.93 | 0.04 | -0.17 | 0.01 | 0.03 | |
| | *Moraxellaceae Acinetobacter* | 0.00% | 0.00% | 0.38% | -0.41 | 1.93 | 0.01 | 9.66 | 5.38 | 0.01 | |
| | *Pseudomonadaceae Pseudomonas* | 0.00% | 0.00% | 0.12% | -1.71 | 0.20 | 0.04 | 8.17 | 5.22 | 0.02 | |
| | *Proteobacteria Unclassified* | 0.04% | 0.01% | 0.03% | -1.53 | 5.2 | 0.047 | 0.84 | 2.0 | 0.060 | |
| | *Micrococcaceae Rothia* | 0.05% | 0.02% | 0.02% | -1.49 | 3.22 | 0.04 | 0.04 | 0.08 | 0.03 | |

**Fig 5. Taxonomic changes grouped by phylum and direction of change in relative abundance that were significantly different in either time point comparison by PERMANOVA analysis.** Taxa with an FDR corrected p-value of >0.05 at either time point were omitted, apart from 1 which was not significant by PERMANOVA but constituted a large proportion of the microbiome indicated by **. Log2 fold change (Log2FC) between mean relative abundances at each time point are colored in orange for increasing fold change, and blue for decreasing fold change. Shading in the variable important to projection (VIP) column indicates the most important variables to the sPLS-DA projections in Fig 2 with darker shading indicating higher cumulative VIP scores. Trend arrows indicate the direction of change in abundance between time points; bent arrows show the direction of change between the first and second time points, followed by the change from the second to third time points. Additional data for species not included above is available in S1 Table in S3 File.

the small sample size, high variability, and semi quantitative nature of non-targeted metabolite profiling, metabolite data was omitted from this analysis.

Between 4 and 9 months of age, most species of Actinobacteria and Proteobacteria decreased in relative abundance, replaced by increasing relative abundance of species of Firmicutes. An apparent increase in relative abundance of Bacteroidetes was not significant. Within the Firmicutes phylum, the 4 to 9 months of age comparison was characterised by significant increases in sequences assigned to species belonging to the *Ruminococcaceae*, *Lachnospiraceae*,

| K pathways level 1 & 2 | | KEGG pathways level 3 | 4 months | 9 months | 12 months | Log2 FC | VIP Sum | fdr p-value | Log2FC | VIP Sum | fdr p-value | Trend |
|---|---|---|---|---|---|---|---|---|---|---|---|---|
| Brite Hierarchies | Protein families: signaling and cellular processes | ko04812 Cytoskeleton proteins | 0.25% | 0.32% | 0.34% | 0.36 | 5.32 | 0.00 | 0.06 | 0.57 | 0.00 | |
| | | ko01504 Antimicrobial resistance genes | 0.36% | 0.28% | 0.26% | -0.34 | 0.51 | 0.01 | -0.11 | 2.40 | 0.01 | |
| | | ko02044 Secretion system | 1.35% | 0.89% | 0.85% | -0.61 | 0.71 | 0.00 | -0.06 | 0.09 | 0.00 | |
| | | ko02035 Bacterial motility proteins | 0.50% | 0.30% | 0.29% | -0.73 | 0.54 | 0.02 | -0.06 | 0.20 | 0.01 | |
| | | ko02000 Transporters | 11.19% | 9.99% | 9.51% | -0.16 | 4.64 | 0.00 | -0.07 | 6.09 | 0.00 | |
| | Protein families: metabolism | ko00194 Photosynthesis proteins | 0.25% | 0.31% | 0.34% | 0.30 | 4.01 | 0.00 | 0.13 | 7.05 | 0.00 | |
| | | ko01008 Polyketide biosynthesis proteins | 0.10% | 0.05% | 0.03% | -1.09 | 0.09 | 0.01 | -0.69 | 1.42 | 0.01 | |
| | | ko01005 Lipopolysaccharide bios. proteins | 0.43% | 0.35% | 0.29% | -0.33 | 0.34 | 0.03 | -0.24 | 2.54 | 0.03 | |
| | | ko01001 Protein kinases | 0.65% | 0.50% | 0.50% | -0.39 | 7.13 | 0.00 | 0.00 | 0.64 | 0.00 | |
| | | ko01009 Protein phosphatases & proteins | 0.12% | 0.16% | 0.16% | 0.34 | 7.28 | 0.00 | -0.01 | 0.17 | 0.00 | |
| Cellular community - prokaryotes | | ko05111 Biofilm formation- *Vibrio cholerae* | 0.33% | 0.21% | 0.17% | -0.61 | 0.32 | 0.00 | -0.30 | 5.50 | 0.00 | |
| | | ko02026 Biofilm formation- *Escherichia coli* | 0.51% | 0.35% | 0.34% | -0.55 | 2.32 | 0.00 | -0.02 | 0.11 | 0.00 | |
| | | ko02025 Biofilm formation- *Pseudomonas aerugin* | 0.25% | 0.15% | 0.14% | -0.75 | 8.05 | 0.00 | -0.12 | 0.77 | 0.00 | |
| Environmental Information Processing | | ko04151 PI3K-Akt signaling pathway | 0.02% | 0.03% | 0.04% | 0.79 | 0.83 | 0.00 | 0.17 | 2.15 | 0.00 | |
| | | ko02020 Two-component system | 1.74% | 1.36% | 1.34% | -0.35 | 0.43 | 0.02 | -0.03 | 1.73 | 0.03 | |
| | | ko02060 Phosphotransferase system (PTS) | 0.72% | 0.55% | 0.49% | -0.39 | 0.15 | 0.03 | -0.17 | 0.37 | 0.03 | |
| Genetic Information Processing | | ko03013 RNA transport | 0.02% | 0.04% | 0.04% | 0.55 | 1.75 | 0.02 | 0.24 | 0.70 | 0.03 | |
| | | ko03000 Transcription factors | 1.51% | 1.13% | 1.12% | -0.41 | 5.52 | 0.00 | -0.02 | 0.24 | 0.00 | |
| | | ko04141 Protein processing in ER | 0.04% | 0.05% | 0.05% | 0.43 | 2.49 | 0.00 | -0.03 | 0.39 | 0.01 | |
| Metabolism | Xenobiotics | ko00627 Aminobenzoate degradation | 0.04% | 0.02% | 0.02% | -0.67 | 0.16 | 0.00 | -0.14 | 0.20 | 0.01 | |
| | | ko00362 Benzoate degradation | 0.10% | 0.06% | 0.06% | -0.69 | 0.09 | 0.03 | -0.01 | 0.19 | 0.04 | |
| | | ko00980 Metabolism by cytochrome P450 | 0.05% | 0.03% | 0.03% | -0.72 | 9.98 | 0.00 | -0.24 | 2.20 | 0.00 | |
| | | ko00982 Drug metabolism - cytochrome P450 | 0.05% | 0.03% | 0.03% | -0.72 | 9.94 | 0.00 | -0.25 | 2.31 | 0.00 | |
| | | ko00930 Caprolactam degradation | 0.02% | 0.01% | 0.01% | -1.38 | 0.98 | 0.00 | -0.22 | 0.04 | 0.00 | |
| | Terpenoids and polyketides | ko01053 Biosynthesis of siderophore nr peptides | 0.12% | 0.05% | 0.04% | -1.17 | 0.02 | 0.00 | -0.56 | 2.66 | 0.00 | |
| | | ko00281 Geraniol degradation | 0.04% | 0.01% | 0.01% | -1.68 | 2.85 | 0.00 | -0.46 | 0.68 | 0.00 | |
| | | ko00903 Limonene and pinene degradation | 0.05% | 0.03% | 0.03% | -0.52 | 4.36 | 0.00 | -0.24 | 2.22 | 0.00 | |
| | Cofactors and vitamins | ko00860 Porphyrin and chlorophyll metabolism | 0.36% | 0.47% | 0.48% | 0.38 | 3.22 | 0.03 | 0.03 | 1.80 | 0.03 | |
| | | ko00130 Ubiquinone & terpenoid-quinone bios. | 0.19% | 0.14% | 0.10% | -0.43 | 0.15 | 0.00 | -0.40 | 8.56 | 0.00 | |
| | | ko00830 Retinol metabolism | 0.04% | 0.03% | 0.02% | -0.45 | 2.36 | 0.00 | -0.21 | 1.01 | 0.00 | |
| | Lipid metabolism | ko00121 Secondary bile acid biosynthesis | 0.05% | 0.06% | 0.07% | 0.44 | 5.28 | 0.00 | 0.13 | 0.81 | 0.00 | |
| | | ko01040 Biosynthesis of unsaturated fatty acids | 0.03% | 0.02% | 0.02% | -0.57 | 2.75 | 0.00 | -0.04 | 0.78 | 0.00 | |
| | Glycan | ko00540 Lipopolysaccharide biosynthesis | 0.27% | 0.18% | 0.14% | -0.54 | 0.30 | 0.02 | -0.43 | 2.99 | 0.03 | |
| | Energy | ko00195 Photosynthesis | 0.25% | 0.31% | 0.34% | 0.31 | 4.07 | 0.00 | 0.13 | 7.15 | 0.00 | |
| | Carbohydrate | ko00053 Ascorbate and aldarate metabolism | 0.20% | 0.16% | 0.14% | -0.32 | 0.19 | 0.01 | -0.18 | 2.88 | 0.00 | |
| | Secondary metabolite bios | ko00966 Glucosinolate biosynthesis | 0.05% | 0.06% | 0.06% | 0.33 | 0.14 | 0.00 | 0.10 | 2.73 | 0.00 | |
| | | ko00333 Prodigiosin biosynthesis | 0.06% | 0.07% | 0.07% | 0.30 | 0.79 | 0.01 | 0.05 | 0.64 | 0.00 | |
| | Amino acid | ko00310 Lysine degradation | 0.24% | 0.17% | 0.15% | -0.51 | 0.60 | 0.00 | -0.21 | 7.33 | 0.00 | |
| | | ko00380 Tryptophan metabolism | 0.18% | 0.12% | 0.11% | -0.57 | 6.11 | 0.00 | -0.22 | 6.30 | 0.00 | |
| | | ko00480 Glutathione metabolism | 0.40% | 0.32% | 0.29% | -0.30 | 6.33 | 0.00 | -0.15 | 2.64 | 0.00 | |
| | | ko00410 beta-Alanine metabolism | 0.14% | 0.11% | 0.10% | -0.37 | 0.25 | 0.02 | -0.18 | 0.91 | 0.02 | |
| | | ko00440 Phosphonate & phosphinate metabolism | 0.07% | 0.05% | 0.05% | -0.53 | 0.55 | 0.03 | -0.04 | 3.74 | 0.03 | |

**Fig 6. Changes in KEGG pathway assignments over time.** Grouping is by parent pathways. The table is filtered to include KEGG pathway assignments that were significantly different in either time point comparison by PERMANOVA analysis and had a log2 fold change >|0.2|. Log2 fold change (Log2FC) between mean relative abundances at each time point are colored in orange for increasing fold change intensity, and blue for decreasing fold change intensity. Shading in the variable important to projection (VIP) column indicates the most important variables to the sPLS-DA projections in Fig 2 with darker shading indicating higher cumulative VIP scores. Trend arrows indicate the direction of change in abundance between time; bent arrows show the direction of change between the first and second time points, followed by the change from the second to third time points. Additional data for species not included above is available in S2 Table in S3 File.

*Peptostreptococcaceae, Clostridiales, Veillonellaceae,* and *Erysipelotrichaceae* families (Fig 5). Some species of *Ruminococcus, Erysipelotoclostridium, Bacillaceae,* and *Christensenellaceae* decreased in relative abundance at 9 months of age, before recovering and surpassing prior abundances at 12 months of age, which strongly influenced the separation of clustering between 9 and 12 months of age in sPLS-DA (Fig 5). With the exceptions of *Clostridium neonatale,* which was not detected at 12 months of age, and *Clostridium sensu stricto,* species belonging to the Firmicutes family were detected at higher relative abundance at 12 months of age compared to at 4 months of age (Fig 5). Populations of some *Peptostreptococcaceae* subspecies and *Veillonellaceae* increased significantly during the 4 to 9 month of age comparison before decreasing at 12 months of age (Fig 5).

Most species of Actinobacteria and Proteobacteria significantly decreased over time. However, the apparent decrease in the most abundant taxonomy assignment, *Bifidobacteriaceae,*

was not significant by PERMANOVA, and this remained the single most abundant taxonomic assignment at 12 months of age (Fig 5). Whereas some *Gammaproteobacteria*, including the *Enterobacteriaceae* family and *Vibrionaceae vibrio*, decreased consistently and significantly over time, other Gammaproteobacteria such as *Moraxcellaceae* and *Pseudomonas* increased in relative abundance after a decrease between 4 and 9 months of age (Fig 5).

More KEGG pathway assignments (57%) were significant by PERMANOVA analysis than Metaxa2 assignments (39%) for at least one time point comparison. The magnitude of log2 fold-change of sequences assigned to KEGG pathways was smaller than the fold-change of sequences assigned to taxonomy (Figs 5 and 6), in line with patterns identified in Fig 3.

The magnitude of fold-change was greater between 4 and 9 months of age than between 9 and 12 months of age (Fig 6), in agreement with clustering patterns shown in Fig 4. More sequences assigned to KEGG pathways significantly decreased in relative abundance between 4 and 9 months of age (Fig 6), also in agreement with diversity scores shown in Fig 3 and clustering identified in Fig 4.

Sequences assigned to "amino acid metabolism", "xenobiotic degradation and metabolism", and "terpenoid and polyketoid metabolism" consistently decreased over time, along with associated "protein families: metabolism" pathways (Fig 6). In particular, decreased cytochrome P450 degradation pathways had a strong influence on the separation between 4 and 9 months of age sPLS-DA clustering. Sequences assigned to "secondary metabolite biosynthesis" and "secondary bile acid biosynthesis" increased consistently in relative abundance over time, whereas "biosynthesis of lipopolysaccharides" and associated "protein family: metabolism" pathways decreased in relative abundance (Fig 6). Of the cellular processes that changed significantly and with a log2 fold change of $>|.2|$, sequences assigned to the "PI3k-Akt signalling pathway" and "RNA transport" increased significantly over time (Fig 6). Sequences assigned to "biofilm formation" from specific species all significantly decreased between each time point (Fig 6).

## Correlation and integration of the microbiome with metabolites between time points

Regularised canonical correlations based on Pearson's correlations of the delta (change) between time points of microbiome datasets (taxonomic assignments (level 6), KEGG pathway assignments (L3)), and metabolome datasets were carried out to identify significantly correlated shifts between time points. Correlations between changes in relative abundances of KEGG pathways and changes in metabolite concentration had more significant coefficients than correlations between microbial species and fecal metabolites.

Filtered heatmaps depicting correlations above the cut-off values are shown in Figs 5–8.

There were more significant correlations, both positive and negative, between 4 and 9 months of age than between 9 and 12 months of age. These observations are in agreement with the results depicted in Figs 3, 5, 6. However, of the taxonomy and KEGG pathway assignments that were found to correlate significantly with metabolites, few overlapped with assignments found to be significant by PERMANOVA shown in Figs 5 and 6 in the 4 vs 9-month time point comparison. Sequences assigned to KEGG pathways correlated more strongly with aqueous metabolites than lipid metabolites, whereas microbial species correlated more strongly with more lipid metabolites. This pattern was particularly apparent between 9 and 12 months of age (Figs 6 and 7).

Between 4 and 9 months of age Unclassified Firmicutes, Unclassified Gammaproteobacteria, and *Enterobacteriales* were the only taxonomic assignments that correlated strongly with metabolites and changed significantly (PERMANOVA FDR corrected p-value of $<0.05$; Figs 5

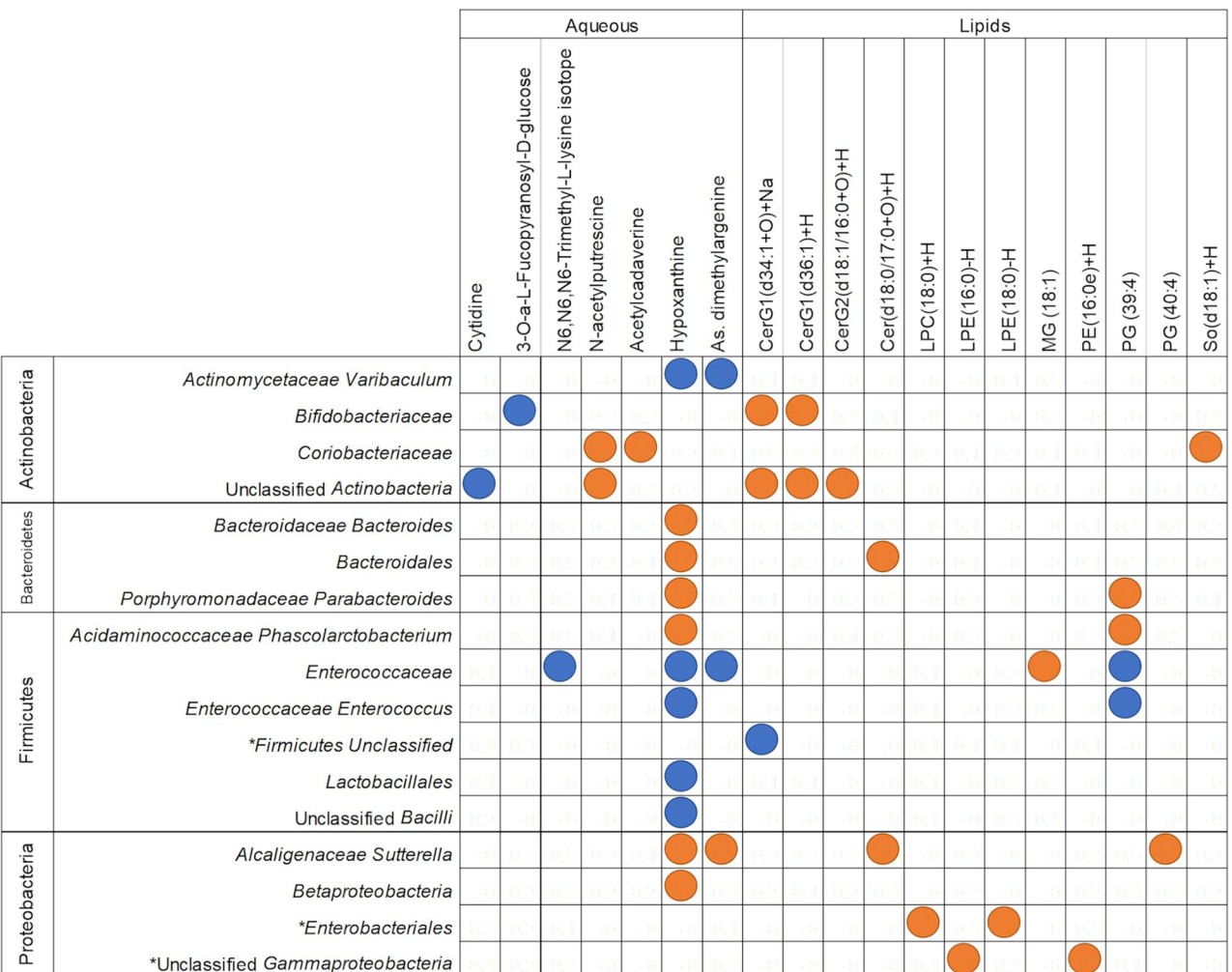

**Fig 7. Filtered regularised canonical correlation heatmap of the delta value of taxonomic assignments and metabolites between 4 and 9 months of age.** Orange circles depict significant positive Pearson's correlation coefficients (>0.5), and blue depicts significant negative correlations (<-0.5). Taxonomic assignments that were also significant by PERMANOVA analysis for this time point comparison are indicated by *.
CerG1 = Glucosylceramide; CerG2 = Diglycosylceramide; Cer = Ceramide; LPC = Lysophosphatidylcholine; LPE = Lysophosphatidylethanolamine; MG = Monoglyceride; PE = Phosphatidylethanolamine; PG = Phosphatidylglycerol; SO = Sphingosine.

and 7). Of the sequences assigned to KEGG pathways that correlated strongly with changes in metabolite concentrations, the "PI3K-Akt signalling pathway" and "RNA transport" pathway assignment increased significantly, and "benzoate degradation" pathway assignments decreased significantly (PERMANOVA FDR corrected p-value of <0.05) (Figs 6 and 8). The remaining sequences assigned to taxonomy and KEGG pathways that correlated with metabolites were not significantly different between 4 and 9 months of age by PERMANOVA (Figs 5 and 6).

Significant correlations were found between changes in sequences assigned to each of the four main phyla and changes in both lipid and aqueous metabolites between 4 and 9 months of age (Fig 7). Hypoxanthine was strongly correlated with members of each phylum; positively with Proteobacteria and Bacteroides, and negatively with Actinobacteria (*Actinomycetaceae Varibaculum)* and several species of Firmicutes (Fig 7). Hypoxanthine was also positively

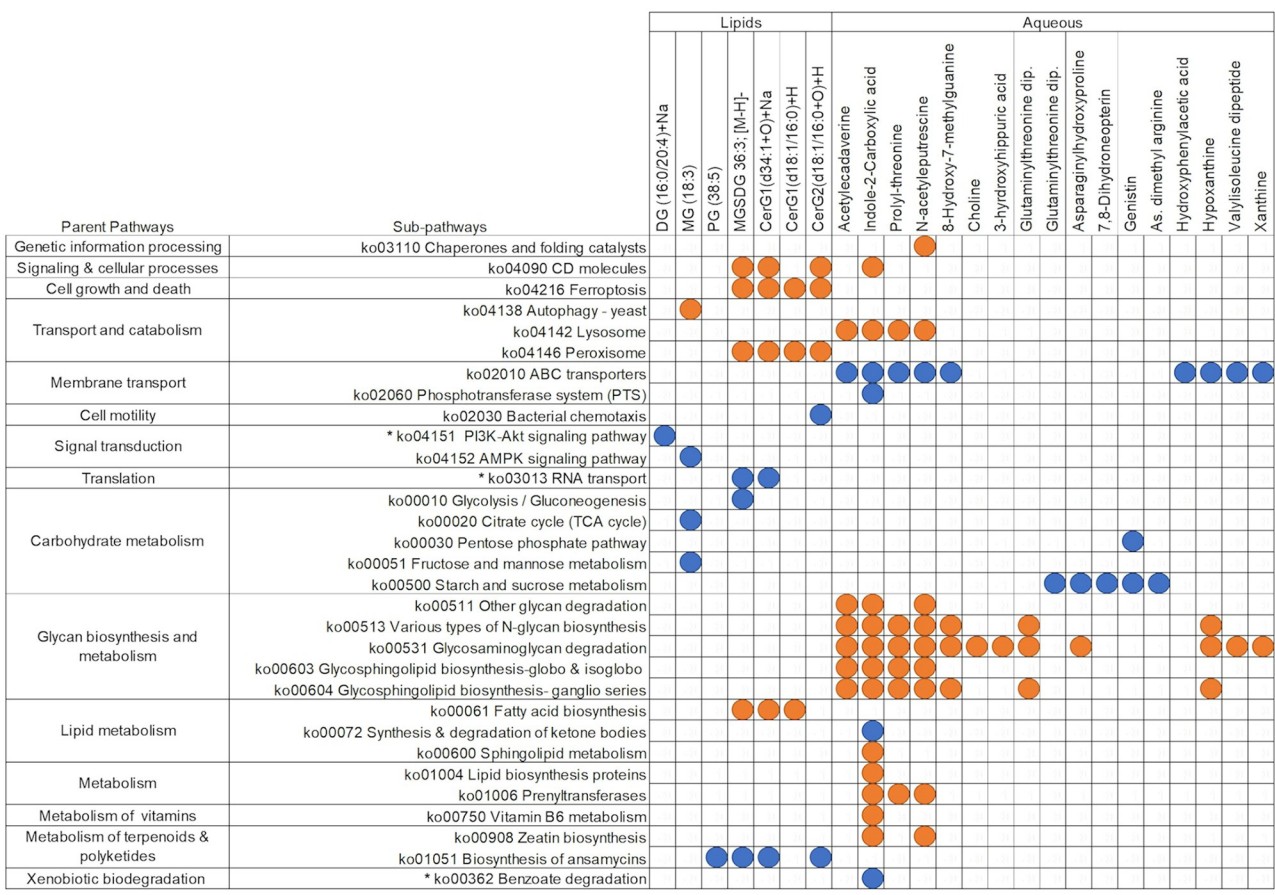

**Fig 8. Filtered regularised canonical correlation heatmap of the delta value of KEGG pathway assignments and metabolites between 4 and 9 months of age.** Orange circles depict significant positive Pearson's correlation coefficients (>0.6), and blue depicts significant negative correlations (<-0.6). KEGG pathway assignments that were also significant by PERMANOVA analysis for this time point comparison are indicated by *.
DG = Diglyceride; MG = Monoglyceride;PG = Phosphoglyceride; PG = Phosphatidylglycerol; MGSDG = Monogalactosyldiacylglycerol;
CerG1 = Glucosylceramide; CerG2 = Diglycosylceramide.

correlated with glycan biosynthesis and metabolism in the integration with KEGG pathway assignments (Fig 7). This positive correlation was part of a larger cluster of aqueous metabolites that positively correlate with sequences assigned to "glycan degradation and biosynthesis," which otherwise did not strongly correlate with any lipid metabolites (Fig 8). Microbial "Carbohydrate metabolism" was negatively correlated with lipid metabolites, and microbial "starch and sucrose metabolism" was negatively correlated with a series of aqueous metabolites (Fig 8).

Between 9 and 12 months of age, fewer aqueous metabolites significantly correlated with taxonomic assignments than with KEGG pathways (Figs 6 and 7). Changes in several classes of lipid metabolites correlated with changes in sequences assigned to species (Fig 9). Firmicutes and Proteobacteria were strongly positively correlated with ceramides, diglycerides, and a triglyceride, but negatively associated with cytidine, an aqueous metabolite (Fig 9). *Clostridium sensu stricto* and both species of *Lachnospiraceae* included here were negatively correlated with lipid metabolites (Fig 9).

A series of dipeptides and amino acids were associated with sequences assigned to KEGG pathways between 9 and 12 months of age (Fig 10). Whereas others were negatively associated

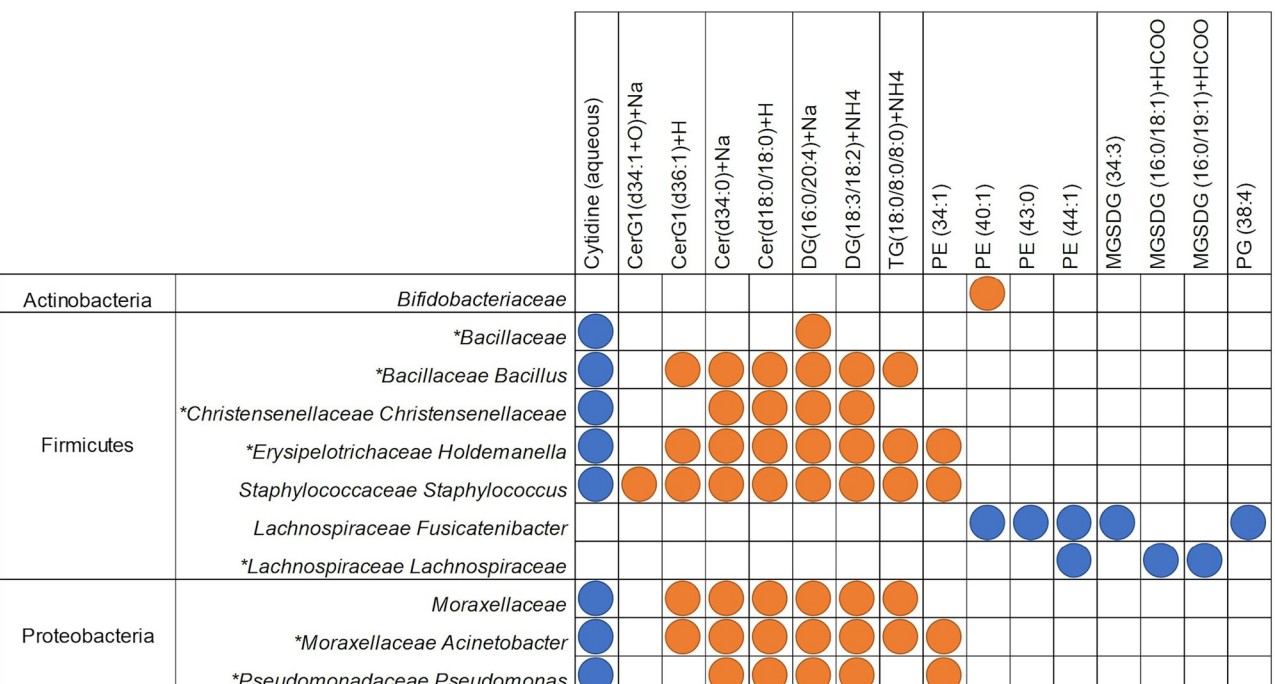

**Fig 9. Filtered regularised canonical correlation heatmap of the delta value of taxonomic assignments and metabolites between 9 and 12 months of age.** Orange circles depict significant positive Pearson's correlation coefficients (>0.5), and blue depicts significant negative correlations (<-0.5). Taxonomic assignments that were also significant by PERMANOVA analysis for this time point comparison are indicated by *. Cer = Ceramide; CerG1 = Glucosylceramide; DG = Diglyceride; TG = Triglyceride; PE = Phosphatidylethanolamine; MGSDG = Monogalactosyldiacylglycerol; PG = Phosphatidylglycerol.

with serine in particular: "biosynthesis of unsaturated fatty acids," "selenocompound metabolism," "protein kinases," "transporters," and "quorum sensing" were positively associated with dipeptide metabolites, "citrate cycle," "metabolism of cofactors and vitamins," "transcription machinery," and "AMPK signalling" (Fig 10).

## Discussion

This study investigated how the infant gut microbiome adapts to the addition and continuation of solid foods during the first year of life by characterising shifts in species composition and functional capacity and identifying significant correlations with gut metabolites. Fecal samples were utilized as a proxy of the microbiome and metabolome in the lower gut. The novel aspect of this study was the identification of metabolites that associate with composition and predictive functional capacity of the microbiota before and after the introduction of solid foods. These highly correlated metabolites may be relevant compounds for further investigations into complementary feeding using targeted metabolomics.

The infant gut microbiome increased in alpha diversity when complementary foods were introduced from after 4 months of age until 12 months of age. In agreement with this study's hypothesis, the shifts occurring between 4 and 9 months of age were more significant than between 9 and 12 months of age. A greater proportion of sequences assigned to KEGG pathways were significant by PERMANOVA than sequences assigned to taxonomy, indicating more consistent changes in functional capacity than in species composition. However, the magnitude of changes in the relative abundance of sequences assigned to functional capacity

**Fig 10. Filtered regularised canonical correlation heatmap of the delta value of KO pathway assignments and metabolites between 9 and 12 months of age.** Orange circles depict significant positive Pearson's correlation coefficients (>0.6), and blue depicts significant negative correlations (<-0.6). KEGG pathway assignments that were also significant by PERMANOVA analysis for this time point comparison are indicated by *.

was smaller than that of changes in microbial composition over the same period. This pattern indicates a degree of functional redundancy across gut microbial species, in particular genes that are compulsory for survival, and suggests that shifts in functional capacity patterns are similar among infants as the diet transitions from milk to solid foods.

The higher variance (beta diversity) of both taxonomy and KEGG pathway assignments at 4 months of age compared to 9 or 12 months of age suggests that the effects of early-life factors with known impacts on the infant gut microbiome that differed among infants in this cohort diminished with the addition of solid foods. Effects of mode of birth on the gut microbiome, including increased skin associated species and functional diversity in Caesarian-born infants, have been reported to persist until approximately 6 months of age [2], but the magnitude and persistence of these influences may vary. The mode of milk feeding has been shown to affect the composition of the gut microbiome, but as time progresses, differences between breastmilk or formula consumption diminish [23]. Individual microbial differences that may have been associated with these early-life factors may become abrogated with the addition of solid foods, as suggested by the tighter clustering of taxonomic and functional assignments in ordination analyses at 9 months of age. These findings are in agreement with published studies [24, 25]. The variance in the gut microbiome composition at 12 months of age may be reflective of microbes' preferences for specific structures from the food matrix that is likely diversifying and divergent between infants as they age.

The patterns that appeared through correlating the change in microbial sequences assigned to species or KEGG pathways with the change in fecal metabolite concentrations support

results from PERMANOVA and ordination analyses, i.e., that more significant changes occurred with the introduction of solid foods rather than the continuation of solid foods. Interestingly, few of the taxa and functional pathways that correlate strongly with metabolites were significant by PERMANOVA analysis, particularly from 4 to 9 months of age. This result suggests that the change in the relative abundance of sequences assigned to taxa or KEGG pathways shown in Figs 5–8 (that were not also significant by PERMANOVA) were more closely related to changes in metabolite concentrations than to age.

The propensity for taxa to correlate more strongly with lipid metabolites than aqueous metabolites suggest these taxa may have a direct or indirect role in the production of these lipids or that the microbes themselves have a higher content of these lipids than other members of the community [26]. The increased variation in sequences assigned to taxonomy and in lipid metabolite profiles between infants at 9 and 12 compared to at 4 months of age supports the theory that the lipids identified were from the bacteria themselves. The correlations between differences in aqueous metabolites and differences in metabolism and degradation KEGG pathway assignments implicate shifting dietary patterns and altered utilisation of substrates derived from the digestion of dietary proteins, saccharides, oligosaccharides, and fermentative by-products from other microbes [27].

Many of the correlated aqueous metabolites detected are products of incomplete protein degradation and putrefaction [28]. This series of dipeptides, along with acetylcadaverine and N-acetylputrescine, are negatively correlated with sequences assigned to "starch and sucrose metabolism," indicating that as protein utilisation decreases, as shown in the decreased amino acid utilisation pathways over time, carbohydrate utilisation increases, which may reflect the protein to carbohydrate ratio in breastmilk and formula [29]. Prior research has identified the gut microbiome as playing a critical role in the production of essential amino acids that are critical to infant growth and development, particularly in the context of breastmilk and formula consumption [1]. The enrichment of amino acid utilisation and biosynthetic pathways at 4 months of age compared to 9 or 12 months of age may compensate for immature proteolytic enzyme production and secretion in the developing gut [1, 30]. The incomplete digestion of proteins can have a significant effect on microbial trophic networks, which may influence immune system development through the production of precursors and intermediaries to immune crosstalk throughout the gut [24, 31].

As trophic networks adapt to shifting resources, interactions between populations of microbial community members also flux, likely in response to metabolite intermediaries. For instance, the strong positive correlation between change in serine concentration, and changes in sequences assigned to quorum sensing pathways, in which acyl-homoserine lactone is an intercellular signalling molecule in gram-negative bacteria, points to the role of metabolites in the regulation of microbial genes that monitor and respond to population density [32, 33]. Small and insignificant (FDR corrected p-value of >0.5) Apparent decreases in sequences assigned to quorum sensing pathways were identified by PERMANOVA but these were not significant (S2 Table in S3 File). This inconsistent change in KEGG pathway abundance over time among infants alongside a positive correlation with serine concentration changes indicate that changes in serine concentrations may have a stronger influence than age in regulating genes associated with monitoring population density.

This study was limited by the small cohort size (n = 25), varied age at sampling, analysing samples in 2 batches, and alterations in sample processing protocols for samples collected at 4 months of age. There were additional limitations inherent in using shotgun metagenomic sequencing to assign taxonomy and in annotating chemical identities to chromatographic peaks during metabolomic data processing [34, 35]. The small size of the Nourish to Flourish cohort limits the confidence with which early-life factors such as mode of birth and mode of

milk feeding can be investigated. However, the design of this study allowed for the effects of the introduction of solid foods on the gut microbiome to be compared with the effects of the continuation of solid foods, using a longitudinal, high-resolution, multi-omic approach. Batch effects were successfully overcome by including samples from all time points in both batches during sample analysis, and normalisation approaches specific to each dataset. Furthermore, additional dietary data were collected to accompany the samples used here, enabling further research into specific associations between complementary feeding and the development of the infant gut microbiome.

The Nourish to Flourish longitudinal study provided evidence that the infant gut microbiome responds to the introduction of solid foods with increased microbial taxonomic diversity while the abundance of functional genes remains relatively stable. Ongoing dietary diversification may lead to continued changes in the relative abundances of different species during the transition from liquid to solid diets. The decreased magnitude of changes in the functional pathway assignments of the gut microbiome from 9 to 12 months of age was indicative of the ongoing stabilisation process. Some changes in relative abundances of microbes and functional groups were more strongly associated with changes in metabolite concentrations than with changes in age. Overall, the lesser changes in gene functions over time, despite large changes in taxonomic composition, suggests that most of the genes required to make use of new substrates are already present at 4 months of age. It is plausible that the changing environment at that time results in the selection of different microbes that may harbour these genes in different combinations.

### Abbreviations and file naming

CF1 refers to data obtained from samples collected at approximately 4 months of age. CF3 refers to data obtained from samples collected at 9 months of age. CF6 refers to data obtained from samples collected at 12 months of age. Documents with titles beginning with KEGG refer to relative abundance data, intermediary analyses, processing, and formatting of KEGG orthologous pathways. Documents with titles beginning with MBComp refer to relative abundance data, intermediary analyses, processing, and formatting of sequences assigned species data. Documents with titles beginning with HILIC refer to aqueous metabolite peak data, intermediary analyses, processing, and formatting of metabolites detected by HILIC LC/MS data. Documents with titles beginning with Lipid refer to lipid peak data, intermediary analyses, processing, and formatting of Lipid LC/MS data. As comparisons were made between timepoints, files containing "BM" refer to comparisons between 4 and 9 months of age, and files containing "ME" refer to comparisons between 9 and 12 months of age. sPLSDA stands for sparse Partial Least Squared Discriminant Analysis. VIP refers to Variable Important to Projection. CIM refers to canonical correlation matrices.

### Supporting information

**S1 File.**
(7Z)

**S2 File.**
(ZIP)

**S3 File.**
(DOCX)

**S1 Checklist. CONSORT 2010 checklist of information to include when reporting a pilot or feasibility trial**\*.
(DOC)

**S1 Protocol. Study protocol—Feasibility study.**
(DOCX)

## Acknowledgments

We thank Olivier Gasser (Victoria University, Wellington New Zealand) for providing immunological expertise to the study; Hedley Stirrat for providing LC-MS technician support and training; and APC Mrobiome Ireland—Teagasc for carrying out metagenomic sequencing.

## Author Contributions

**Conceptualization:** Martin Kussman, Clare R. Wall.

**Data curation:** Starin McKeen, Hannah Eriksen, Amy Lovell, Wayne Young, Karl Fraser.

**Formal analysis:** Starin McKeen.

**Funding acquisition:** Nicole C. Roy, Martin Kussman, Clare R. Wall, Warren C. McNabb.

**Investigation:** Starin McKeen, Hannah Eriksen, Amy Lovell.

**Methodology:** Starin McKeen, Nicole C. Roy, Jane Adair Mullaney, Hannah Eriksen, Amy Lovell, Wayne Young, Karl Fraser, Clare R. Wall, Warren C. McNabb.

**Project administration:** Clare R. Wall, Warren C. McNabb.

**Resources:** Wayne Young, Karl Fraser.

**Software:** Jane Adair Mullaney, Wayne Young, Karl Fraser.

**Supervision:** Jane Adair Mullaney, Wayne Young, Karl Fraser, Warren C. McNabb.

**Validation:** Jane Adair Mullaney, Hannah Eriksen, Amy Lovell, Wayne Young, Karl Fraser.

**Visualization:** Starin McKeen.

**Writing – original draft:** Starin McKeen.

**Writing – review & editing:** Nicole C. Roy, Jane Adair Mullaney, Hannah Eriksen, Amy Lovell, Wayne Young, Karl Fraser, Clare R. Wall, Warren C. McNabb.

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
