## [Decision Letter · Decision Letter 0]

6 Dec 2021

PONE-D-21-23857Adaptation of the infant gut microbiome during the complementary feeding transitionPLOS ONE

Dear Professor  McNabb,

Thank you for submitting your manuscript to PLOS ONE. After careful consideration, we feel that it has merit but does not fully meet PLOS ONE’s publication criteria as it currently stands. Therefore, we invite you to submit a revised version of the manuscript that addresses the points raised during the review process.

This is an interesting study. However,  this longitudinal and multiple omics study was designed with small sample sizes and the data was analyzed using some inappropriate models or methods.  After reading this manuscript I agree with these two reviewers' comments on its limitations of design and methods used. Please address each points made by these two reviewers and the editor and especially the suggestions by this editor about the analyses.

We look forward to receiving your revised manuscript.

Kind regards,

Yinglin Xia, Ph.D.

Academic Editor

PLOS ONE

2. For more information on PLOS ONE's expectations for statistical reporting, please see https://journals.plos.org/plosone/s/submission-guidelines.#loc-statistical-reporting. Please update your Methods and Results sections accordingly.

“No”

6. We note that you have indicated that data from this study are available upon request. PLOS only allows data to be available upon request if there are legal or ethical restrictions on sharing data publicly. For information on unacceptable data access restrictions, please see http://journals.plos.org/plosone/s/data-availability#loc-unacceptable-data-access-restrictions.

Additional Editor Comments:

1. This is longitudinal study design. However cross-sectional models were used for almost analyses, Such as the t-test between sampling time points for sequences. This is not correct.

2. Was the data tested for normality before choosing tests or models?

3. Levels 1 to 4 KEGG were analyzed. More information about these levels of KEGG such as what they are about should be described to improve readability.

4. Since this is clinical trial, did the study design was based on power analysis? In other words, the used samples sizes were based on power calculations? How many percentage of power for this study? What model or test was used for the power calculation?

5. Table 1. Taxonomic compositional changes over time at phylum level are not meaningful. The taxa at the lower levels such as genus or species are required to be analyzed.

6. The microbiome data were normalized using the total sum scaling. However, the total sum scaling is the simplest normalization method and was evaluated by several studies as not optimal for microbiome data.

Reviewers' comments:

Reviewer's Responses to Questions

**Comments to the Author**

1. Is the manuscript technically sound, and do the data support the conclusions?

Reviewer #1: No

Reviewer #2: Partly

2. Has the statistical analysis been performed appropriately and rigorously? 

Reviewer #1: No

Reviewer #2: No

3. Have the authors made all data underlying the findings in their manuscript fully available?

Reviewer #1: Yes

Reviewer #2: Yes

4. Is the manuscript presented in an intelligible fashion and written in standard English?

Reviewer #1: Yes

Reviewer #2: Yes

5. Review Comments to the Author

Reviewer #1: Major Questions/Comments:

1. In Fig. 1, it shows clearly n=40 infants were randomized by sex (as stated in line 101) into goups of n=30 and n=10. Thus this will lead to a higly unbalanced comparison. In addition, the final comparison is based on (valid data of) n=20 versus n=5. Personally I think it's hard to draw reasonal statistical conclusions based on n=5.

2. Correlation cutoffs 0.6/0.5 were used for KEGG pathway/species correlations (as mentioned in line 215/216). I understand different cutoffs for different correlations could be used. But some rationale are expected to be mentioned, i.e., why to use 0.6/0.5 as the cutoffs for KEGG pathway/species correlations?

Minor Comments:

1. Extra "is a" at line 73.

2. At line 577, do we mean "Small and insignificant (FDR corrected p-value >0.05) appreant...", i.e., should be 0.05 instead of 0.5 for the "cutoff" for p-values?

Reviewer #2: 1. Sample size/power calculation should be performed when design the study. If not. clarify this is a pilot study.

2. For a small sample size, it is hard to determine whether the data follows a normal distribution. Nonparametric method may be considered (e.g. Spearman correlation and Friedman test). Or consider data transformation.

3. Little details were discussed for the two intervention groups. It does not serve the research aims and no comparisons between the intervention were performed.

4. Fig 3AB. What method was used for the confidence bands?

5. Fig 3 CD. T-test is not appropriate. Consider PERMANOVA/Friedman test.

6. PLOS authors have the option to publish the peer review history of their article (what does this mean?). If published, this will include your full peer review and any attached files.

Reviewer #1: No

Reviewer #2: No

---

## [Author Response · Author response to Decision Letter 0]

17 May 2022

Thank you for your helpful comments on our manuscript. 

1. In line with the journal requirements, we have amended the manuscript to comply with PLOS ONE style from the templates indicated at 

2. Have updated our methods and results sections

3. Have made sure the grant and funding information match

4. Have completed the competing interests 

5. Have made the data available as additional supplementary tables

https://mfr.osf.io/render?url=https%3A%2F%2Fosf.io%2F3zfre%2Fdownload

6. See 5.

7. Included captions for supporting information files at the end of the manuscript and updated in text citations to match accordingly.

Our response to the comments from the reviewers

1. This pilot study was a longitudinal design and multivariate analyses were applied and when ANOVA analysis identified significance, post-hoc testing was done, which were the ‘T tests’ referred to. It is appropriate to apply post-hoc testing if ANOVA finds significance.

2. We applied a permutation ANOVA to the data. Permutation ANOVA is a test that does not require gaussianity. It is therefore robust and resistant to skewed distribution of the data and does not need normality tests prior.

3. We have added more information about the KEGG levels to the manuscript line 140-143.

4. There was no power analysis as this reports the findings from a pilot study which had the purpose of seeing whether this type of study involving infants and a dietary intervention were feasible. There was no primary endpoint and the two groups were either those who included a probiotic or those who added NZ Kumara to the diet of the infants.

5. Taxonomic compositional changes over time at phylum level are meaningful when dealing with an infant undergoing weaning. We know that profound changes to the microbiome occur during the period of introduction to solids. We found that there is a large shift towards taxa of the Firmicutes phylum and away from taxa from the Actinobobacteria phyla when the diet was expanded to include solids. 

6. We agree that we could have used alternative normalization strategies for microbiome data. However, infants have a unique situation that they are changing rapidly as they grow and one of the challenges that we hoped to overcome was the ability to measure changes over time that were not confounded by the physical changes in environment space also expanding. The microbiome increases with the infant growth and so we found that for this study, normalization through total sum scaling and then converting to abundance, allowed us to compare the same infant across time. 

In response to whether we describe a technically sound piece of scientific research:

The study we report on was designed as a pilot to see what challenges and difficulties there might be in an infant clinical trial were we to use a dietary intervention. For a pilot study, there was no clinical endpoint measured and there was no power analysis required. 

We have endeavoured to better explain the reasons and rationale for the methods used and the conclusions made. We did not carry out T tests as such, but post-hoc tests from ANOVA if the results of this indicated that there was a significant (P<0.05) difference between groups. Permutation ANOVA also is a robust method not requiring the test for normality of data.

We used total sum scaling and abundance because of the count changes in the number of bacteria enumerated as the infants aged from 3-12 months. We used total sum scaled taxonomy and metagenome datasets, and log-transformed metabolome data also because these methods maintain positive values, which is critical when assessing the direction of change in abundance or concentration.

1. In Fig. 1, it shows clearly n=40 infants were randomized by sex (as stated in line 101) into goups of n=30 and n=10. Thus this will lead to a higly unbalanced comparison. In addition, the final comparison is based on (valid data of) n=20 versus n=5. Personally I think it's hard to draw reasonal statistical conclusions based on n=5.

Response: We acknowledge that this was a pilot study that had 40 recruited (also minimal numbers) and then we were only able to use 25. We agree with the reviewer; it is hard to draw statistical conclusions on n=5. As a pilot study we can report the results we do have so long as we do not overinterpret the results. 

2. Correlation cutoffs 0.6/0.5 were used for KEGG pathway/species correlations (as mentioned in line 215/216). I understand different cutoffs for different correlations could be used. But some rationale are expected to be mentioned, i.e., why to use 0.6/0.5 as the cutoffs for KEGG pathway/species correlations?

Response: We advise that we used a 0.6 cut off for the KEGG pathway correlations because the networks identified are presumed to be a stronger correlation. The 0.5 cut off was used for the species correlations to help identify potential pathways for further investigation.

Minor Comments:

1. Extra "is a" at line 73.

Response: corrected

2. At line 577, do we mean "Small and insignificant (FDR corrected p-value >0.05) appreant...", i.e., should be 0.05 instead of 0.5 for the "cutoff" for p-values?

Response: Correct it should say 0.05 and we have amended the manuscript.

Reviewer #2: 1. Sample size/power calculation should be performed when design the study. If not. clarify this is a pilot study.

Response: We confirm that it is a pilot study as mentioned in the abstract and described at line 69 where we wrote “This is a secondary analysis of a pilot study. The pilot study was designed to evaluate the methodologies required for a randomized control trial (RCT).”

3. For a small sample size, it is hard to determine whether the data follows a normal distribution. Nonparametric method may be considered (e.g. Spearman correlation and Friedman test). Or consider data transformation. 

Response: Permutation ANOVAs were carried out and they use non-parametric methods.

4. Little details were discussed for the two intervention groups. It does not serve the research aims and no comparisons between the intervention were performed.

Response: This is correct as it was a pilot study. We did not find any statistically significant differences between the dietary groups and so we reported that changes over time were the key findings.

5. Fig 3AB. What method was used for the confidence bands?

Response: These are not confidence bands. The wording has been amended to describe this figure better. They indicate minimum-maximum values.

6. Fig 3 CD. T-test is not appropriate. Consider PERMANOVA/Friedman test.

Response: Post-hoc testing was carried out only after ANOVA identified significant differences between the groups. We have amended the wording to reflect this.

---

## [Editor Report · Decision Letter 1]

7 Jun 2022

Adaptation of the infant gut microbiome during the complementary feeding transition

PONE-D-21-23857R1

Dear Dr. Mullaney,

We’re pleased to inform you that your manuscript has been judged scientifically suitable for publication and will be formally accepted for publication once it meets all outstanding technical requirements.

Kind regards,

yinglin xia, Ph.D.

Academic Editor

PLOS ONE
---

## [Editor Report · Acceptance letter]

20 Jun 2022

PONE-D-21-23857R1 

Adaptation of the infant gut microbiome during the complementary feeding transition 

Dear Dr. Mullaney:

I'm pleased to inform you that your manuscript has been deemed suitable for publication in PLOS ONE. Congratulations! Your manuscript is now with our production department. 

Kind regards, 

on behalf of

Dr. yinglin xia 

Academic Editor

PLOS ONE